# Is price associated with the quality of medicines? Evidence from active pharmaceutical ingredient testing in Nigeria

Marie Chantel Montás[1]*, Chimezie Anyakora[2,3], Elisa Maria Maffioli[4]

1 Department of Global Health and Population, Harvard T. H. Chan School of Public Health, Boston, Massachusetts, United States of America, 2 Bloom Public Health, Asokoro, Abuja, Nigeria, 3 School of Science and Technology, Pan Atlantic University, Lagos, Nigeria, 4 Department of Health Management and Policy, School of Public Health, University of Michigan, Ann Arbor, Michigan, United States of America

* mariemontas@fas.harvard.edu

## Abstract

Determining the quality of medicines remains a challenge, particularly in low- and middle-income countries, where regulatory oversight and enforcement vary, and resources and infrastructure for quality testing are often constrained. In these settings, price is often used as a proxy for higher-quality medicines, yet empirical evidence supporting this assumption remains scarce. We conducted a mystery shopper survey in over 1,200 retail pharmacies across urban and rural areas in the six geopolitical zones of Nigeria, purchasing one drug sample from a list of twenty branded medicines, including analgesics, antimalarials, antibiotics, antihypertensives, and multivitamins. A sub-sample of the purchased medicines (N = 246) was tested for quality, defined as passing a laboratory test using High-Performance Liquid Chromatography (HPLC) to measure the Active Pharmaceutical Ingredient (API) content of each medicine. Using probit regressions, we examined the extent to which price is associated with quality, controlling for observable pharmacy and drug sample characteristics. A 1% increase in price is associated with a 16.7 percentage point increase in the probability of passing the laboratory test, conditional on other factors. Receiver Operating Characteristic (ROC) analysis shows strong out-of-sample classification performance, with an Area Under the Curve (AUC) of 0.82 for the price-only model, indicating that price alone explains much of the variation in quality. Other results show that medicines organized by brand and displaying visible expiration dates may signal higher quality, while the presence of other observable characteristics (e.g., packaging, storage, display) shows more counterintuitive associations with drug quality in this context. Stratified analyses show that the association between price and quality is particularly strong for analgesics and antibiotics. These findings suggest that price appears to be a reliable signal of medicine quality, whereas other characteristics of pharmacies and drug samples provide weaker and less consistent indicators. This underscores the need for stronger regulatory oversight, greater market

**Data availability statement:** Laboratory results for the subsample of tested medicines (N=246) are available in the Supplementary Material of the following publication: Maffioli EM and Anyakora C. "A Comparative Study Between Near-Infrared (NIR) Spectrometer and High-Performance Liquid Chromatography (HPLC) on the Sensitivity and Specificity." PLOS ONE. 2025. DOI: https://doi.org/10.1371/journal.pone.0319523.

**Funding:** The project was funded by USAID-DIV (7200AA21FA00006) (EM) and internal funding from the University of Michigan (EM). The funders did not play any role in the study design, data collection and analysis, decision to publish, or preparation of the manuscript.

**Competing interests:** The authors have declared that no competing interests exist.

transparency, and targeted consumer education to promote safer access to quality medicines.

## Introduction

Substandard and falsified (SF) medicines pose a serious threat to global health, particularly in low- and middle-income countries (LMICs), where an estimated 10.5% of medicines are of poor quality, contributing to around 1 million deaths annually [1]. Their prevalence varies widely by region and drug type, with some studies indicating that up to 50% of medicines in certain low-income settings, especially in Sub-Saharan Africa, are SF [2–6]. A recent systematic review indicates the prevalence of SF medicines in Sub-Saharan Africa to be 22.6% [7]. Antibiotics, antimalarials, and antihypertensives are the most affected drugs, drawing significant concern [7–9].

Beyond their direct impact on patient health—causing adverse effects, treatment failure, and preventable deaths [5,10]—SF medicines also fuel the rise of drug resistance [11], undermining efforts to control infectious diseases. The consequences of SF medicines extend beyond healthcare, placing additional financial burdens on individuals and health systems, eroding trust in medical institutions, and exacerbating poverty through income loss and reduced productivity. Ultimately, the widespread presence of SF medicines not only threatens individual well-being but also weakens economies and health systems.

SF medicines are difficult to detect as they often closely resemble genuine products in packaging and appearance, lack visible quality indicators, and require advanced laboratory testing to verify their chemical composition. The illegal trafficking of counterfeit medicines is a highly profitable business, valued between US$ 200 billion and US$ 431 billion, making it the most lucrative sector in the global trade of counterfeit goods [12]. In 2010, the United Nations Office on Drugs and Crime identified for the first time the trafficking of SF medical products as a substantial threat, alongside cocaine, maritime piracy, and human trafficking. In LMICs, the challenge of regulating SF medicines is exacerbated by weak drug registration systems and inadequate enforcement of quality standards [13]. Additionally, corruption and institutional weaknesses often undermine government efforts to address inefficiencies in the drug supply chain, while financial constraints limit individuals' choices, making them more vulnerable to purchasing poor-quality medicines from informal sources.

In an imperfect pharmaceutical market, consumers often lack complete information about medicine quality, making it difficult to distinguish between SF medicines and those of verified quality. Various factors influence perceived quality, with price being one of those [14–16]. In fact, given quality uncertainty and significant price variations across pharmacies, consumers may rely on market indicators, price being one of them, as a heuristic proxy for quality [14]. The challenge of assessing medicine quality is particularly relevant in LMICs, where pharmaceutical expenditures account for 20% to 60% of total health spending, compared to just 18% in Organization for Economic Co-operation and Development countries [1]. In LMICs, limited access to health insurance forces most individuals to pay out-of-pocket for medications. Thus,

financial constraints may push economically disadvantaged consumers to choose lower-priced medicines, hoping they will be as much as effective, despite concerns about quality.

Beyond price, observable indicators such as pharmacy characteristics, including air conditioning, refrigeration for temperature-sensitive medicines, and proper storage to prevent exposure to direct sunlight, can serve as indirect proxies of quality. Similarly, drug packaging features, such as intact seals, undamaged pills, and visible expiration dates, may influence consumer perceptions of quality. Since direct measures of drug quality are difficult to obtain for consumers, such indirect signals – pharmacy characteristics, drug sample features (including packaging attributes – e.g., box size, type of cardboard, text clarity on packaging and blister packs), and price – may be used to assess perceived quality [17]. This study explored the extent to which consumers could rely on these observable indicators for pharmacies and drug packaging characteristics, as well as price, as signals of quality of medicines.

The study was conducted in Nigeria, where the pharmaceutical market is valued at US$ 4.5 billion and growing at a rate of 9% annually [18]. The country is highly import-dependent, sourcing 70% of its finished products from abroad and relying almost entirely on other countries for active pharmaceutical ingredients (APIs) needed for local manufacturing [19,20]. Analgesics hold the largest market share (25%), followed by antibiotics (15%), multivitamins (15%), antimalarials (14%), and antihypertensives (8%) [21]. The prevalence of SF drugs in Nigeria is estimated between 4.95% and 25% [5,8]. Since the early 2000s, the National Agency for Food and Drug Administration and Control (NAFDAC) has aggressively fought against SF medicines through legislative changes, new guidelines, stricter enforcement, the dismissal of corrupt personnel, audits, the destruction of large quantities of SF products, and public awareness campaigns, leading to a substantial reduction in their prevalence [22]. More recently, NAFDAC has gained international recognition for its use of cutting-edge technologies such as Raman Spectroscopy, the GPHF Minilab, and Mobile Authentication Service. In January 2024, it launched the "Green Book" to verify medicines [23].

Our study relied on data we gathered in November 2022 from more than 1,200 retail pharmacies across urban and rural locations in the largest cities of Nigeria's six geopolitical zones. Mystery shoppers purchased one drug from a list of twenty branded medicines across five categories: analgesics, antimalarials, antibiotics, antihypertensives, and multivitamins, while also recording observable pharmacy and drug sample characteristics. A sub-sample of these medicines (N = 246) underwent quality testing, between November 2022 and February 2023, through a quantitative analysis of APIs using High-Performance Liquid Chromatography (HPLC).

This study contributes to the existing evidence by providing empirical insights into the association between price and quality of medicines in a market such as Nigeria, where SF medicines pose a significant health risk. While previous research has documented the prevalence and consequences of SF medicines, there is limited quantitative evidence on how consumers navigate quality uncertainty and whether price may serve as a reliable indicator of the quality of medicines. This study quantitatively assesses the extent to which price is associated with quality as measured by an acceptable API content (90–110% range) using HPLC, controlling for other indicators, including pharmacy and drug sample characteristics. It aims to inform policy interventions aimed at strengthening consumer protection and pharmaceutical regulation in LMICs.

## Materials and methods

### Study setting and design

During a three-week period from November 10th to the 30th of 2022, we purchased drug samples from randomly selected retail pharmacies in both rural and urban locations across the largest cities in Nigeria's six geopolitical zones: Abuja, Kano, Lagos, Onitsha, Port Harcourt, and Yola [S1 Fig], [S2 Fig]. We followed a standardized sampling strategy in which enumerators started at a designated (recorded) location within an urban or rural area of the city and then conducted random walks to identify the next pharmacy. A team of 12 enumerators from Bloom Public Health acted as mystery shoppers, purchasing a randomly selected branded drug from a list of twenty medicines at each pharmacy visited (see [24] for the full list). Shoppers underwent thorough training and were provided with a script outlining a cover story for each

drug purchase. They were instructed to either retrieve the medication from the counter or interact with the pharmacist to request it if necessary. Two enumerators were assigned to each urban and rural area in the six cities in the study. Enumerators followed a random-walk sampling approach, beginning from a chosen starting point in their city and recording GPS coordinates for verification. Each visited approximately 7–13 pharmacies per day, purchasing one medicine per pharmacy without revealing the study purpose or negotiating prices. These methods using mystery shoppers or simulated patients along with random walking are commonly employed in pharmacy research [25–27]. This study was approved by the Research Ethics Committee on Human Subjects at the University of Michigan (HUM00214684) and the Health Research Ethics Committee (HREC) in Nigeria (NHREC/01/01/2007). This study does not entail human subject research, and verbal or written consent does not apply.

The twenty selected drugs spanned categories including analgesics, antimalarials, antibiotics, antihypertensives, and multivitamins, representing medicines with the highest market share, as reported by one of our local partners. None of these medicines required cold-chain storage, and all were stable at room temperature under standard storage conditions (below 30°C and away from direct sunlight). We aimed to purchase around 200 drugs per city, for a total of almost 1,200 drug samples, evenly split between rural and urban areas. This way, we stratified our sample by city and geographical area. In 93.7% of the visits, the enumerators successfully found the first drug on the list (N = 1,214).

After each visit to a pharmacy, the enumerators filled out a survey to gather observed characteristics of the pharmacy and the drug samples purchased. Because directly measuring consumer access to information on the quality of medicines is challenging, we identified indirect indicators as rough proxies. There are various ways in which drug quality can be signaled, and with the survey we assessed the extent to which consumers could potentially infer the quality of medicines from pharmacy and drug sample characteristics, or the price. Enumerators also took a photo of the pharmacy for visual confirmation of their visit. The survey tool can be found in [S4 Table].

A sub-sample of the drugs (N = 246) was sent to the laboratory for testing using HPLC to measure the API content of each medicine. HPLC analysis was conducted at Hydrochrom Analytical Services Limited, a registered private laboratory (RC-1864978) located in Gowon Estate, Lagos, between November 2022 and February 2023. Drug samples were initially collected and categorized at the research team's partner office in Abuja before being shipped to the laboratory for analysis (more information on the testing procedures can be found in [24]). Additional methodological details are provided in the supplementary information of [24].

This statistically representative sub-sample was selected using a weighted average by type of drugs of the total 1,124 purchased medicines, excluding multivitamins, which were uncommon in the pharmacies sampled and would not allow for enough variation for analysis (see [24] for more information). When tested with HPLC, 25% (N = 62) failed due to API falling outside the 90–110% range, indicating a high prevalence of poor-quality medicines in our context. Of those failing the HPLC test, 35% (N = 22) were antihypertensives, 31% (N = 19) were analgesics, 19% (N = 12) were antibiotics, and 15% (N = 9) were antimalarials [5] (see supplementary information in [24]).

## Empirical model

We explored the factors potentially associated with the quality of medicines as measured by an acceptable API content (90–110% range) as tested by HPLC (we refer to this measure of quality hereafter): (i) *price,* measured in natural logarithms to account for the left-skewed nature of prices and to express results in terms of price elasticity; (ii) *characteristics of pharmacies*, such as presence of air conditioning, drugs displayed on shelves, and cool-chain devices; (iii) *characteristics of drug samples*, such as whether the storage temperature is listed on the package, the condition of the packaging, whether the drugs are organized by brand, and if the expiration date is visible on the package; and, (iv) covariates, including the category of medicine (analgesics, antibiotics, antihypertensives, antimalarials), the location of the pharmacy (city or rural/urban area), the size of the pharmacy, categorized as small (1–2 vendors), medium (3–4 vendors), or large (5 or more vendors), and the origin of the manufacturer (Nigerian or international).

Our empirical model is a probit regression, as follows:

$$PassLAB_{icl} = \gamma_1 ln\,(price)_{icl} + \gamma_2 pharma_{icl} + \gamma_3 drug_{icl} + X_{icl} + \varepsilon_{icl} \tag{1}$$

where $i$ refers to the drug sample purchased at the pharmacy by the mystery shopper and tested at the laboratory, in category $c$ and location $l$. Our outcome of interest, $PassLAB_{icl}$ is a binary variable indicating whether the drug passes the HPLC test. The regressors of interest include $ln(price)_{icl}$, the natural logarithm of price; $pharma_{icl}$, a vector of characteristics of the pharmacy where the drug sample was purchased; $drug_{icl}$, a vector of characteristics of the drug samples, and $X_{icl}$, a vector of covariates as described above. Standard errors are clustered at the city level. Our hypothesis is that $\gamma 1 > 0$, meaning that price is positively associated with the quality of medicines. We will test whether this is true, conditional on the characteristics of the pharmacy, drug samples, and other covariates. Our estimates are expressed as marginal effects, which should be interpreted as a percentage change in the probability of passing the laboratory test (i.e., having API content within acceptable 90–110% range) in response to a 1% change in price. All quantitative analyses were performed using StataNow 19.5 MP-Parallel Edition, and all figures were generated in RStudio (Version 2023.03.0+386).

## Results

### Sample characteristics

Table 1, column 1 presents summary statistics for the full sample of 1,214 medicines purchased across the pharmacies. The data confirm that the sample was evenly distributed across the six cities, with approximately 16.31% to 17.13% of observations in each city, and an almost equal split between urban (49.59%) and rural (50.41%) areas, consistent with the targeted sample. Pharmacy size, proxied by the number of vendors, shows that most pharmacies (69.19%) were small, with 1–2 vendors. Medium-sized pharmacies (3–4 vendors) accounted for 20.10%, while large pharmacies (5 or more vendors) represented 10.71%. Finally, most of the manufacturers of the drugs purchased were Nigerian, accounting for 65.35% of the total sample.

Regarding pharmacy characteristics, enumerators observed that 94.15% of pharmacies displayed medicines on shelves, and in 7.41% of pharmacies the medicines were directly exposed to sunlight. Additionally, 8.26% of pharmacies had medicines placed on the floor. 35.54% of the pharmacies had air conditioning, and 26.36% had cool-chain devices for medicines requiring constant refrigeration (e.g., for oxytocin).

For drug sample characteristics, 98.85% of the medicines purchased had all pills present, 98.11% had packaging in normal condition (not broken, cracked, or scratched), 98.27% had intact pills, and 92.50% had a visible expiration date on the package. The average price in the sample was NGN 1,660.8 (US$3.79), with prices ranging from NGN 250 (US$0.57) to NGN 7,500 (US$ 17.1). Antihypertensives were the most expensive at NGN 2,462 (US$ 5.61), while analgesics were the least expensive at NGN 1398.4 (US$ 3.19). The characteristics of the pharmacies where the sub-sample of drugs tested in the laboratory were purchased (N=246) are similar to those of the full sample [Table 1, column 2], reflecting the random sampling approach.

### Price is associated with the quality of medicines

To examine whether price varies by our measure of quality of medicines, we first plotted the density distribution of price (measured in natural logarithm, ln) for drug samples that passed and failed the laboratory test [Fig 1]. We found that the distributions are bimodal but differ by sub-sample. Overall, the average price (ln) of drug samples that fail the laboratory test (ln 7) is statistically significantly lower than that of those samples that did not failed (ln 7.3) (two-sided p-value=0.02, 95%CI [−0.37, −0.03]). However, the distribution of drug samples that passed the laboratory test is more skewed to the right than that of the samples that failed. This may suggest a positive association between price and quality of medicines as measured by having API content within acceptable 90–110% range. Additionally, an asymptotic two-sided

**Table 1. Characteristics of pharmacies and medicines.**

| | (1)<br>Mean (%) (SD)<br>[Min-Max]<br>Full sample<br>N = 1,214 | (2)<br>Mean (%) (SD)<br>[Min-Max]<br>Tested in the laboratory<br>N = 246 |
|---|---|---|
| **Panel A: Characteristics of pharmacies** | | |
| Air conditioning | 35.34 (47.82) | 31.3 (46.67) |
| Medicines displayed on shelves | 94.15 (23.48) | 95.93 (19.79) |
| Cool-chained devices | 26.36 (44.08) | 26.02 (43.96) |
| Medicines exposed to direct sunlight | 7.41 (26.20) | 7.32 (26.09) |
| Medicines placed on the floor | 8.26 (27.53) | 8.94 (28.59) |
| **Panel B: Characteristics of drug samples** | | |
| Package in normal condition | 98.11 (13.64) | 97.15 (16.66) |
| Expiration date on package | 92.50 (26.34) | 92.68 (26.09) |
| Storage temperature on package | 82.37 (38.12) | 82.11 (38.40) |
| Medicines organized by brand | 72.73 (44.55) | 73.98 (43.96) |
| Drug sample passed the laboratory test (i.e., API content within 90–110% range) | – | 74.80 (43.50) |
| **Panel C: Other covariates** | | |
| *City* | | |
| Abuja | 16.72 (37.33) | 15.45 (36.21) |
| Kano | 16.31 (36.96) | 15.04 (35.82) |
| Lagos | 17.13 (37.7) | 15.45 (36.21) |
| Onitsha | 16.47 (37.11) | 18.70 (39.07) |
| Port Harcourt | 17.05 (37.62) | 19.11 (39.39) |
| Yola | 16.31 (36.96) | 16.26 (36.98) |
| *Geographical area* | | |
| Urban | 49.59 (50) | 48.78 (50.09) |
| Rural | 50.41 (50) | 51.22 (50.09) |
| *Size of vendor* | | |
| Small (1–2 vendors) | 69.19 (46.19) | 71.54 (45.21) |
| Medium (3–4 vendors) | 20.10 (40.09) | 16.26 (36.98) |
| Large (5 or more vendors) | 10.71 (30.93) | 12.20 (32.79) |
| *Category of drug samples* | | |
| Analgesics | 30.72 (46.15) | 44.72 (49.82) |
| Antimalarials | 30.4 (46.02) | 27.24 (44.61) |
| Antibiotics | 18.29 (38.67) | 15.45 (36.21) |
| Antihypertensives | 15.57 (36.27) | 12.60 (33.25) |
| *Manufacturer (Nigerian)* | 65.35 (47.60) | 64.63 (47.90) |
| *Price* | | |
| Price (Naira) | NGN 1660.8, USD 3.79 (NGN 894.3, USD 2.04)<br>[NGN 250USD 0.57 - NGN 7500 USD17.1] | NGN 1598.2, USD 3.64 (NGN 811.39, USD 1.85)<br>[NGN 300, USD 0.68 - NGN 400, USD 0.91] |
| Price – Analgesics | NGN 1398.4, USD 3.19 (NGN 683.8, USD 1.56)<br>[NGN 300, USD 0.68 - NGN 3500, USD 7.98] | NGN 1326.1, USD 3.02 (NGN 638.86, USD 1.46)<br>[NGN 300, USD 0.68 - NGN 3000, USD 6.84] |

*(Continued)*

**Table 1.** (Continued)

| | (1)<br>Mean (%) (SD)<br>[Min-Max]<br>Full sample<br>N = 1,214 | (2)<br>Mean (%) (SD)<br>[Min-Max]<br>Tested in the laboratory<br>N = 246 |
|---|---|---|
| Price – Antibiotics | NGN 1705.8, USD 3.89 (NGN 992.3, USD 2.26)<br>[NGN 300, USD 0.68 - NGN 7500, USD 17.1] | NGN 1700, USD 3.88 (NGN 930.2, USD 2.12)<br>[NGN 450,USD 1.03 - NGN 4000, USD 9.12] |
| Price – Antihypertensives | NGN 2461.8, USD 5.61 (NGN 922.3, USD 2.10)<br>[NGN 750,USD 1.71 - NGN 7500, USD 17.1] | NGN 2280.6, USD 5.20 (NGN 835.33, USD 1.90)<br>[NGN 1000, USD 2.28 - NGN 3500, USD 7.98] |
| Price – Antimalarials | NGN 1662.1, 3.79 (NGN 720.7, USD 1.64)<br>[NGN 250,USD 0.57 - NGN 3000, USD 6.84] | NGN 1671.6, USD 3.81 (NGN 783.55, USD 1.79)<br>[NGN 300, USD 0.68 - NGN 3000, USD 6.84] |

*Notes:* This table reports means as proportions (%), standard deviation (SD), minimum, and maximum for the full sample of drug samples purchased (N = 1,214, column 1) and the sub-sample of drug samples tested by High-Performance Liquid Chromatography (N = 246, column 2). The following variables are summarized: characteristics of pharmacies (Panel A), characteristics of drug samples (Panel B), and other covariates (Panel C), including city, geographical area, category of drug samples, manufacturer type and price (in Nigerian Naira and USD). Exchange rate at November 30, 2022 was Nigerian Naira 438.6 = USD 1 from Nigerian Central Bank: https://www.cbn.gov.ng/rates/ExchRateByCurrency.html.

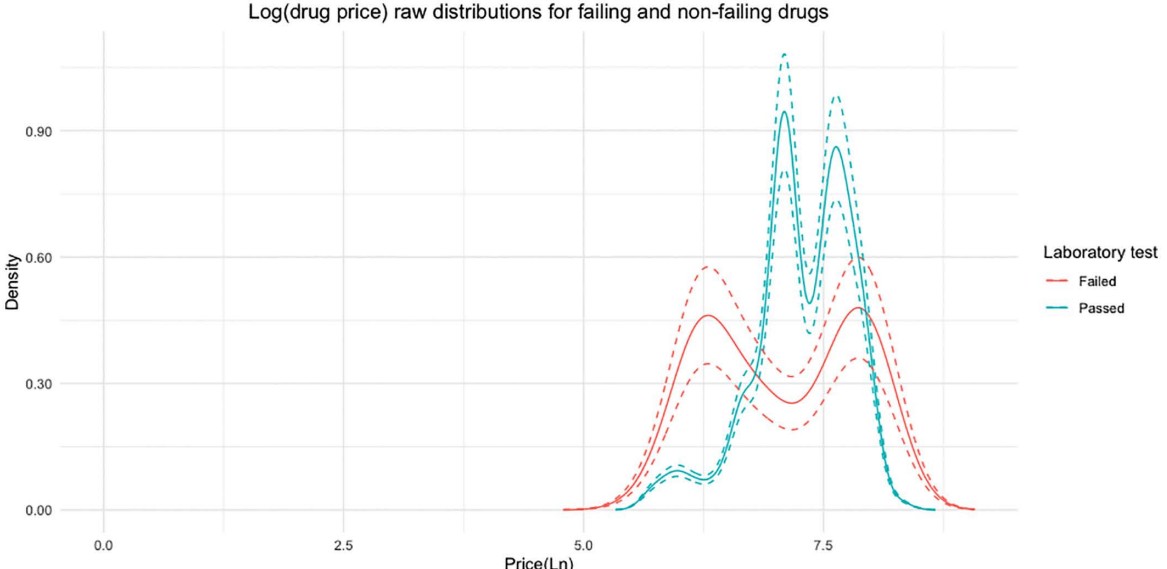

**Fig 1. Density of price of medicines, by pass or fail in the laboratory test.** *Notes:* The figure displays the kernel density estimates of the natural logarithm of price (ln, in Nigerian Naira and USD) for drug samples (N = 246) that either passed or failed the laboratory test. Dotted lines represent the 95% confidence intervals. Exchange rate at November 30, 2022 was Nigerian Naira 438.6 = USD 1 from Nigerian Central Bank: https://www.cbn.gov.ng/rates/ExchRateByCurrency.html.

Kolmogorov-Smirnov test confirms a statistically significant difference at the 99% confidence level in the distribution of price (ln) between drug samples that passed and failed the laboratory test (p-value < 0.001).

Additionally, we plotted the percentage of drug samples across price bins, disaggregated by passing or failing the laboratory test [Fig 2]. This allows for non-parametric comparisons and for observing any potential threshold effects in pricing. The price bins were selected based on the peaks of the linear price distribution in [Fig 1]. The distribution is not uniform across bins: a notably higher proportion of failed drugs fall in the lowest price category ("<NGN 1200 or USD 2.7"), where over half of the drug samples tested failed the laboratory test (51.6%). In contrast, the proportion of passing drugs increases in the mid- to upper-price bins ("NGN 1200–2000 or USD 2.7-4.6", "NGN 2000 or USD 4.6", and ">NGN 2000 or USD 4.6"), suggesting a possible positive association between price and our measure of quality. However, even in the highest price bin (">NGN 2000 or USD 4.6"), a non-negligible fraction of drugs still failed (33.9%), suggesting that high price may not always be a proxy for high quality. These findings suggest that while very low price may signal poor quality, higher price is not a guaranteed signal of high quality.

We further plot the distribution of price by drug samples that passed and failed the laboratory test for each category of medicines [S3 Fig]. The bimodal shape of the distribution is more pronounced for antihypertensives and antimalarials. In contrast, the distributions for analgesics and antibiotics are more spread out, suggesting that price may be more strongly associated with the quality of these two categories of medicines than the quality of antihypertensives and antimalarials. We also observe a similar pattern when plotting the percentage of samples by price bins, for each category of medicines [S4 Fig]. Among analgesics, the failure rate is concentrated in the lowest price bin ("<NGN 1200 or USD 2.7"), where 78.9% of samples failed. This pattern is not observed in the higher price bins. A similarly pronounced pattern is seen for antibiotics, where all failures are confined to the lowest price bin (100%), suggesting a sharp quality threshold at the lower end of the price distribution, which may indicate that low price is a strong signal of poor quality for antibiotics specifically.

Table 2 presents the estimates from our empirical model (1). There is a positive and statistically significant association between price and the quality of medicines at the 1% level across all empirical specifications. This suggests that higher prices are associated with an increased probability of passing the laboratory test, holding other covariates constant (such

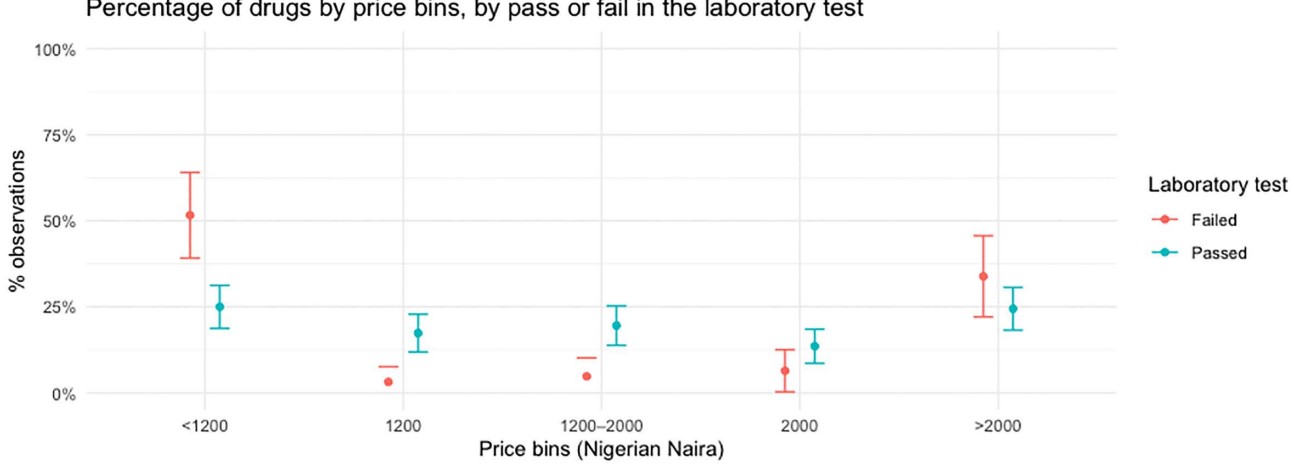

**Fig 2. Percentage of medicines samples by price bins, by pass or fail in the laboratory test.** *Notes:* This figure shows the distribution of laboratory test outcomes across five mutually exclusive price categories: Nigerian Naira <NGN 1200 or USD 2.7, NGN 1200 or USD 2.7, NGN 1200–2000 or USD 2.7 - 4.6, NGN 2000 or USD 4.6, and >NGN 2000 or USD 4.6. Bins lines represent the 95% confidence intervals. Exchange rate at November 30, 2022 was Nigerian Naira 438.6 = USD 1 from Nigerian Central Bank: https://www.cbn.gov.ng/rates/ExchRateByCurrency.html.

**Table 2. Factors associated with quality of medicines.**

| Dependent variable: Passing the laboratory test | | | | |
|---|---|---|---|---|
| | (1) | (2) | (3) | (4) |
| **Price(ln)** | **0.161*** | **0.158*** | **0.170*** | **0.167*** |
| | (0.0510) | (0.0506) | (0.0485) | (0.0487) |
| **Characteristics of pharmacies** | | | | |
| Air conditioning | | −0.0446 | | −0.0835* |
| | | (0.0516) | | (0.0468) |
| Drugs displayed on shelves | | 0.0652 | | −0.103*** |
| | | (0.0400) | | (0.0247) |
| Cool-chained devices | | 0.00281 | | 0.0444 |
| | | (0.0561) | | (0.0511) |
| Medicines exposed to direct sunlight | | 0.143*** | | 0.174*** |
| | | (0.0349) | | (0.0442) |
| Medicines placed on the floor | | −0.0800 | | −0.0848 |
| | | (0.0600) | | (0.0754) |
| **Characteristics of drug samples** | | | | |
| Package in normal condition | | | 0.0536 | 0.0520 |
| | | | (0.158) | (0.134) |
| Expiration date on package | | | 0.140** | 0.246*** |
| | | | (0.0556) | (0.0668) |
| Storage temperature on package | | | −0.0874*** | −0.118*** |
| | | | (0.0272) | (0.0346) |
| Medicines organized by brand | | | 0.109* | 0.135** |
| | | | (0.0588) | (0.0544) |
| **Other covariates** | | | | |
| Kano | 0.0869** | 0.0534 | 0.0525 | 0.00274 |
| | (0.0345) | (0.0474) | (0.0385) | (0.0487) |
| Lagos | 0.166*** | 0.151*** | 0.0673 | 0.00668 |
| | (0.0107) | (0.0295) | (0.0417) | (0.0497) |
| Onitsha | 0.134*** | 0.119*** | 0.0895*** | 0.0405 |
| | (0.0273) | (0.0426) | (0.0181) | (0.0376) |
| Port Harcourt | 0.194** | 0.121 | 0.128 | 0.0193 |
| | (0.0814) | (0.101) | (0.0881) | (0.107) |
| Yola | 0.00589 | −0.00530 | 0.0243** | −0.0121 |
| | (0.0130) | (0.0326) | (0.0122) | (0.0258) |
| Urban | −0.0247 | −0.0235 | −0.0496 | −0.0554 |
| | (0.0594) | (0.0647) | (0.0511) | (0.0528) |
| Medium (3–4 vendors) | −0.0290 | −0.0220 | −0.0273 | −0.0214 |
| | (0.0744) | (0.0726) | (0.0729) | (0.0686) |
| Large (5 or more) | −0.0217 | −0.0240 | −0.0174 | −0.0324 |
| | (0.124) | (0.141) | (0.118) | (0.130) |
| Antibiotics | −0.0973* | −0.106** | −0.114** | −0.131** |
| | (0.0519) | (0.0512) | (0.0575) | (0.0587) |
| Antihypertensives | −0.469*** | −0.446*** | −0.473*** | −0.454*** |
| | (0.0850) | (0.0734) | (0.0902) | (0.0849) |
| Antimalarials | 0.000700 | 0.0114 | −0.00587 | 0.00262 |
| | (0.0390) | (0.0247) | (0.0386) | (0.0253) |

*(Continued)*

**Table 2.** (Continued)

| Dependent variable: Passing the laboratory test | | | | |
|---|---|---|---|---|
| | (1) | (2) | (3) | (4) |
| **Price(ln)** | **0.161***** | **0.158***** | **0.170***** | **0.167***** |
| Nigerian manufacturer | −0.0509 | −0.0559 | −0.0542 | −0.0596* |
| | (0.0359) | (0.0342) | (0.0361) | (0.0336) |
| Mean Dep. Var. | 74.79 | 74.79 | 74.79 | 74.79 |
| Observations | 246 | 246 | 246 | 246 |
| R-squared | 0.257 | 0.263 | 0.275 | 0.287 |

*Notes:* The empirical model is a probit regression, with marginal effects and standard errors exported. The outcome is defined as whether the drug sample passes a laboratory test performed using High-Performance Liquid Chromatography (HPLC). Price is measured in natural logarithm and in Nigerian Naira. Standard errors are clustered and the city level and reported in parentheses. *** $p < 0.01$, ** $p < 0.05$, * $p < 0.1$.

as the category of medicines, location, manufacturer type, and size of the pharmacy, column 1). More specifically, a 1% increase in price is statistically significantly (at 99% confidence level) associated with a 16.1 percentage point increase in the probability of passing the laboratory test. The size of the coefficient remains similar when controlling for pharmacy characteristics (column 2). When controlling for drug sample characteristics (column 3), the magnitude of the coefficient slightly increases to 17.0 percentage points. The estimates are robust when all characteristics are included at 16.7 percentage points (column 4).

When examining the covariates, we find that drug samples from Lagos and Onitsha States are more likely to pass the laboratory test, while those from Yola are less likely to pass, compared to drug samples from Abuja, which is the capital city of Nigeria. Instead, a bigger size of the pharmacy or being in a rural location does not predict our measure of quality. However, antibiotics and antihypertensives are less likely to be of quality compared to analgesics. When all covariates are included—along with pharmacy and drug sample characteristics, as well as price (column 4)—the drug manufacturer being Nigerian is also negatively and statistically significantly correlated (at 90% confidence level) with passing the laboratory test.

In terms of pharmacy characteristics, contrary to our expectations, we find that medicines exposed to direct sunlight are more likely to pass the laboratory quality test (column 2, $p < 0.01$), while no other pharmacy characteristics show a significant association with quality. Although this result may seem counterintuitive, anecdotal reports suggest that higher-quality medicines may be kept more visibly in public-facing positions, such as near windows or displays exposed to sunlight, to avoid detection during audits. Further, when controlling for all observable factors (column 4), medicines displayed on shelves or those purchased from pharmacies with air conditioning are less likely to meet quality standards. One possible explanation is that visually appealing pharmacies may place greater trust in their suppliers or make stocking decisions based on market preferences and aesthetics than on rigorous quality assurance practices. This could explain why medicines displayed on shelves or in air-conditioned pharmacies show lower quality, despite appearing more professional to consumers.

Regarding drug sample characteristics, contrary to our expectations, we find that indicating a temperature on the package is negatively associated with quality (columns 3 and 4). Notably, 18% of samples did not specify a precise storage temperature but included a generic statement such as "do not store above 30°C." This may suggest that certain labeling practices serve more to signal compliance than to reflect actual storage or handling. In contrast, packages that display an expiration date and those from pharmacies that organize medicines by brand are associated with a 14.0 and 10.9 percentage point higher probability, respectively, of passing the laboratory test (column 3). These associations are robust and even stronger in magnitude when controlling for all other observable characteristics (column 4). These features may reflect more rigorous manufacturing standards or inventory management practices.

Taken together, the results suggest that observable characteristics, such as packaging details, display conditions, storage instructions, and organization by brand, do not consistently serve as reliable proxies for medicine quality in the Nigerian context. In several cases, features that might typically indicate proper handling or regulatory compliance—such as being displayed on shelves, kept out of direct sunlight, or including temperature guidance—are associated with lower quality. This counterintuitive pattern may reflect systemic weaknesses in enforcement, where visual cues and superficial compliance mask deeper issues in sourcing and inventory practices.

To further assess the validity of our analyses, we used a Receiver Operating Characteristic (ROC) curve to evaluate the model's ability to distinguish between high-quality and poor-quality drugs across a range of classification thresholds, providing a more comprehensive measure of predictive performance beyond standard goodness-of-fit metrics. The basic receiver operating characteristic (ROC) curve plots the true positive rate (sensitivity) against the false positive rate (1 − specificity) across all possible classification thresholds. In our context, we define a positive case as a high-quality drug, meaning the true positive rate reflects the proportion of actual high-quality drugs correctly identified by the model, while the false positive rate reflects the proportion of poor-quality drugs incorrectly classified as high quality [28].

Fig 3 shows the ROC curves comparison between the price-only model and the full model that includes additional covariates. Each curve shows the trade-off between sensitivity and 1 − specificity for each model. Both models demonstrate strong discriminative ability, with area under the curve (AUC) values of 0.82 (price-only model) and 0.84 (full model). These values suggest that both specifications are effective at distinguishing between drugs that pass or fail the laboratory test. However, the limited difference in model performance from the full model suggests that the price-only model captures a lot of the variation in the laboratory-confirmed quality of medicines. Thus, the incremental benefit of potential factors associated with quality of medicines, in addition to price, may be limited in the context of Nigeria.

Finally, results from related research in [5] indicated that drug samples failed the laboratory test due to either low (26%), high (14%), or both low and high APIs (60%, for samples with two ingredients). We presented robustness checks

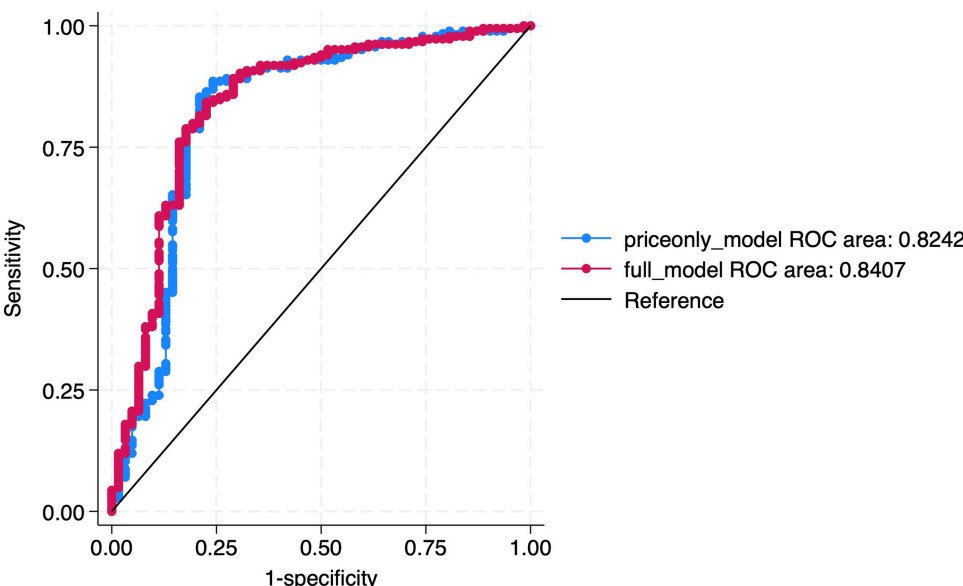

**Fig 3. Receiver Operating Characteristic (ROC) curves for empirical models based on price alone and price combined with other potential factors associated with quality of medicines.** *Notes:* Figure 3 presents ROC analysis used to evaluate out-of-sample classification performance. This approach provides a threshold-independent measure—Area Under the Curve (AUC)—that complements traditional model fit statistics. It highlights the trade-off between sensitivity and specificity, comparing the predictive ability of the baseline and extended models.

exploring the associations between price and quality, as defined by passing the laboratory test based on at least one low API [S1 Table] or at least one high API [S2 Table]. We find evidence that price remains negatively and statistically significantly associated with passing the laboratory test only for drugs with at least one low API [S1 Table, column 4]. We also explored the heterogeneity of the findings by category of medicine and find evidence consistent with [S3 Fig] and [S4 Fig], showing that the stronger positive association between price and our quality of medicines is driven by analgesics and antibiotics (7.89 and 6.62 percentage point, respectively, at 99% confidence interval), conditional on pharmacy, drug sample, and other characteristics [S3 Table]. When examining heterogeneity by medicine category, we use an Ordinary Least Squares (OLS) regression due to convergence issues encountered with the probit specification.

## Discussion

This study investigates the association between pharmacy and drug characteristics and quality of medicines as measured by an acceptable API content (90–110% range) as tested by HPLC, with a particular focus on the role of price in Nigeria's pharmaceutical market. This market is characterized by an annual growth rate exceeding 9% and a significant prevalence of substandard and falsified (SF) drugs, with 25% of tested samples failing this laboratory quality assessment. Our findings indicate that price is strongly associated with this measure of quality of medicines in Nigeria. Specifically, a 1% increase in drug price is associated with a 16.7 percentage point increase in the probability of passing the laboratory test. This highlights the role of price as a primary signal for quality of medicines, even after controlling for proxies of pharmaceutical and drug sample quality, as well as other indicators such as location, pharmacy size, manufacturer type, and category of medicines.

However, relying solely on price as a quality signal may be difficult, especially in markets characterized by imperfect competition and price differentiation [29]. While our results show that very low prices are often associated with poor-quality medicines, they also reveal that price is not a perfect indicator: in our sample, 33.9% of medicines in the highest price bin (>NGN 2000) still failed the quality test. Market variability, informal pricing, and branding may all complicate the association between price and quality for certain medicines.

In such complex environments, consumers may seek to rely on other observable characteristics to infer quality, such as pharmacy or drug sample characteristics. Yet, our findings indicate that these visual indicators are often unreliable. In the Nigerian context, indicators like medicines being organized by brand and having expiration dates visible on the packaging may serve as signals of higher quality. However, several of the associations we observe between observable characteristics and drug quality are shown to be counterintuitive. For instance, indicators that should indicate proper storage, such as medicines displayed on shelves, not exposed to sunlight, or including storage temperature guidance, are instead associated with lower quality. These findings suggest that in Nigeria, consumers may struggle to interpret such cues reliably, despite these features being emphasized in public health messaging as signs of good storage and handling. Taken together, this underscores the central role price may play in shaping consumer perceptions, even when other potentially misleading indicators are present.

Our findings align with limited existing evidence which examined drug samples from various LMICs and found that higher-priced medicines were significantly more likely to pass quality tests, especially in settings with weak regulatory enforcement [3,16]. Higher prices may reflect the greater production costs associated with higher-quality drugs, which typically require superior manufacturing processes, higher-grade active pharmaceutical ingredients (APIs), and compliance with regulatory standards. Additionally, in many LMICs, pharmacies use price to differentiate high-quality drugs from lower-quality alternatives. Some pharmacies sell both legitimate and SF versions of the same medicine, with higher-priced products generally being of better quality [16]. Since drug quality is not directly observable, price can serve as a signal that a product is legitimate and produced by a reputable manufacturer. Our study expands on this limited body of evidence by providing quantitative insights into the association between and price and a measure of quality of medicine in a LMIC context, specifically in Nigeria.

Access to essential medicines remains a global challenge, with many medicines facing high prices, low affordability, and poor availability [30]. Our findings carry important policy implications for addressing these issues. If price reliably signals quality, reducing disparities between high- and low-quality drugs may require improving consumer access to information about observable quality indicators. In LMICs, where financial constraints often limit access to higher-priced drugs, the association between lower price and lower quality presents a significant challenge. Policies aimed at reducing prices of medicines could inadvertently distort consumer perceptions of quality if price could be used as the primary signal. This highlights the need for complementary strategies to enhance transparency in pharmaceutical markets. Strengthening the implementation of regulatory oversight at the local level, expanding routine drug quality monitoring, and promoting authentication technologies could help improve enforcement and empower consumers to make more informed choices. More broadly, the persistent issue of SF medicines in many LMICs underscores the need for continued efforts at both national and international levels. In Nigeria, for example, regulatory initiatives led by NAFDAC, including legislative reforms, staff conduct guidelines, and stricter enforcement of registration and importation standards, have shown promise in improving quality of medicines [31,32]. NAFDAC also achieved WHO Global Benchmarking Tool (GBT) Maturity Level 3 in 2022.

Our research highlights the need to equip consumers and supply chain stakeholders with more reliable information about medicine quality, as most observable pharmacy and drug characteristics we examined do not appear to be reliable signals, at least in the Nigerian context. Reducing reliance on indirect or misleading cues will require more systematic drug testing, transparent reporting of substandard products, and targeted consumer education to improve the identification of poor-quality medicines. Future research should explore interventions that close informational gaps along the supply chain, ultimately empowering consumers and enhancing public health outcomes.

Our study is not without limitations. First, while we establish associations between price and quality of medicines, we cannot infer causal relationships from our results. Second, we identify proxies of quality based on observable characteristics of pharmacies and drug samples collected through mystery shopping. Some may question our classification of these proxies, and we may have overlooked other relevant characteristics not captured in our survey. Third, while our analysis incorporates city-level controls within each geopolitical zone of Nigeria to capture broader spatial variation, we acknowledge that residual neighborhood-level income heterogeneity may influence medicine prices independently of product quality. Future research employing more disaggregated geographical data could better isolate and quantify these localized effects. Finally, our sample is limited to 20 specific branded drugs produced by specific manufacturers and available in Nigeria among four medicine categories. As a result, our findings may not be generalizable to other drug categories or geographical settings.

Despite these limitations, our study fills an important gap by providing quantitative evidence on the association between price and quality of medicines in a rapidly growing LMIC pharmaceutical market such as Nigeria. Our findings highlight the need for policies that balance affordability with quality assurance, ensuring consumers can access safe and effective medicines without needing to rely solely on price as quality indicator.

## Conclusion

In settings with weak regulatory oversight and limited access to product verification, such as Nigeria, consumers often rely on observable cues to assess medicine quality. Yet our findings suggest that many of these cues, including pharmacy and drug sample characteristics, are unreliable or even misleading. Among the available signals, price emerges as the most predictive indicator. This is not because consumers are irrational, but because it conveys more information than other observable factors in this context. These results highlight the urgent need for stronger regulatory enforcement, systematic drug quality testing, and transparent reporting of substandard products. To ensure safer access to medicines, Nigeria must strengthen its regulatory capacity, increase market transparency, and invest in targeted consumer education to help close persistent information gaps.

## Supporting information

**S1 Fig. Location of pharmacies in Nigeria, by geographical area.** *Notes:* National Overview. Geographic distribution of sampled pharmacies across Nigeria, stratified by study state. Each color represents one of the six sampled states. Dots correspond to individual pharmacy locations included in the study. Maps created by the authors using public data from Natural Earth.
(PNG)

**S2 Fig. Location of pharmacies by State, and by geographical area.** *Notes:* Overview by State and Local Governmental Areas. (A) Pharmacy locations in the Federal Capital Territory (FCT). Urban pharmacies (green) are densely concentrated in the city center, with rural pharmacies (orange) more dispersed. (B) Pharmacy locations in Lagos State. Urban pharmacies (green) are clustered around the Lagos metropolitan area, while rural pharmacies (orange) extend eastward and inland. (C) Pharmacy locations in Rivers State. Urban pharmacies (green) are concentrated in the south, while rural pharmacies (orange) are distributed inland. (D) Pharmacy locations in Adamawa State. Urban pharmacies (green) are densely grouped in the central metropolitan zone, with rural pharmacies (orange) forming a surrounding arc. (E) Pharmacy locations in Kano State. Urban pharmacies (green) are clustered in the central part of the state, while rural pharmacies (orange) are more dispersed, extending southward and slightly to the north. (F) Pharmacy locations in Anambra State. The map shows distinct urban clustering (green) in the northern part of the state, with rural sites (orange) scattered throughout. Maps created by the authors using public data from Natural Earth.
(PDF)

**S3 Fig. Density of price of medicines, by pass or fail in the laboratory test, by category of medicines.** *Notes:* The figure shows the kernel densities of price (ln, in Nigerian Naira and in US dollars) for drug samples (n = 246) that passed or failed the laboratory test, by category of medicines: analgesics, antibiotics, antihypertensives and antimalarials. Dotted lines represent the 95% confidence intervals.
(PNG)

**S4 Fig. Percentage of medicines samples by price bins, by pass or fail in the laboratory test, by category of medicines.** *Notes:* This figure shows the distribution of laboratory test outcomes across five mutually exclusive price categories: Nigerian Naira <NGN 1200 or USD 2.7, NGN 1200 or USD 2.7, NGN 1200–2000 or USD 2.7–4.6, NGN 2000 or USD 4.6X, and>NGN 2000 or USD 4.6, by category of medicines: analgesics, antibiotics, antihypertensives and antimalarials. Bins lines represent the 95% confidence intervals. Exchange rate at November 30, 2022 was Nigerian Naira 438.6 = USD 1 from Nigerian Central Bank: https://www.cbn.gov.ng/rates/ExchRateByCurrency.html.
(PNG)

**S1 Table. Factors Associated with Quality of Medicines (Alterative Measure of Quality Based on Low APIs).** *Notes:* The empirical model is a probit regression, with marginal effects and standard errors exported. The outcome is defined as whether the drug non failed a laboratory test performed using High-Performance Liquid Chromatography (HPLC) due to at least a low API (below 99%). Price is measured in natural logarithm and in Nigerian Naira. All empirical models control for location (city and geographical area), drug and manufacturer type, and pharmacy size. Standard errors are clustered at the city level and reported in parentheses. *** p < 0.01, ** p < 0.05, * p < 0.1.
(DOCX)

**S2 Table. Factors Associated with Quality of Medicines (Alterative Measure of Quality Based on High APIs).** *Notes:* The empirical model is a probit regression, with marginal effects and standard errors exported. The outcome is defined as whether the drug non failed a laboratory test performed using High-Performance Liquid Chromatography (HPLC) due to at least a high API (above 110%). Price is measured in natural logarithm and in Nigerian Naira. All empirical models control

for location (city and geographical area), drug and manufacturer type, and pharmacy size. Standard errors are clustered at the city level and reported in parentheses. *** p < 0.01, ** p < 0.05, * p < 0.1.
(DOCX)

**S3 Table. Factors Associated with Quality of Medicines, by Category of Medicine.** *Notes:* The empirical model is an Ordinary Least Squares regression. The outcome is defined as whether the drug non failed a laboratory test performed using High-Performance Liquid Chromatography (HPLC). Price is measured in natural logarithm and in Nigerian Naira. All empirical models control for location (city and geographical area), drug and manufacturer type, and pharmacy size. Standard errors are clustered at the city level and reported in parentheses. *** p < 0.01, ** p < 0.05, * p < 0.1.
(DOCX)

**S4 Table. Survey tool used by enumerators to collect pharmacy- and drug-level data.** *Notes:* This table presents the full survey instrument used by mystery shoppers to collect data during pharmacy visits. The tool includes modules on pharmacy identification, location (urban/rural classification, GPS coordinates), observed storage conditions, and drug sample characteristics (type, dosage, packaging, price, and manufacturer information). Enumerators completed the survey electronically using the KoboCollect App immediately after each purchase to ensure data accuracy and minimize recall bias.
(DOCX)

## Acknowledgments

We thank our partners in Bloom Public Health and Innovations for Poverty Action Nigeria, for purchasing the drugs and sending them to the laboratory for testing. We also thank the 12 enumerators in Bloom Public Health who helped gather the data used in this analysis. Finally, we thank Dr. Felix Khuluza for his valuable suggestions on the manuscript.

## Author contributions

**Conceptualization:** Marie Chantel Montás, Elisa Maria Maffioli, Chimezie Anyakora.

**Data curation:** Marie Chantel Montás.

**Formal analysis:** Marie Chantel Montás.

**Funding acquisition:** Elisa Maria Maffioli, Chimezie Anyakora.

**Investigation:** Marie Chantel Montás, Elisa Maria Maffioli.

**Methodology:** Marie Chantel Montás, Elisa Maria Maffioli.

**Project administration:** Marie Chantel Montás, Elisa Maria Maffioli.

**Resources:** Marie Chantel Montás.

**Software:** Marie Chantel Montás.

**Supervision:** Elisa Maria Maffioli.

**Validation:** Marie Chantel Montás, Elisa Maria Maffioli.

**Visualization:** Marie Chantel Montás.

**Writing – original draft:** Marie Chantel Montás, Elisa Maria Maffioli.

**Writing – review & editing:** Marie Chantel Montás, Elisa Maria Maffioli, Chimezie Anyakora.

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
