## [Decision Letter · Decision Letter 0]

8 Oct 2025

PONE-D-25-41093Is Price Associated with the Quality of Medicines? Quantitative Evidence from NigeriaPLOS ONE

Dear Authors,

Thank you for submitting your manuscript to PLOS ONE. After careful consideration, we feel that it has merit but does not fully meet PLOS ONE’s publication criteria as it currently stands. Therefore, we invite you to submit a revised version of the manuscript that addresses the points raised during the review process.

I have read the paper as well, and I find it interesting and well-written. I align with some of the issues raised by one of the referees. Also, I have some further comments that can help you enhance the paper and unlock its full potential.

Prices of products vary with local income across states as well as across neighborhoods within these states, once you account for neigbhourhood-income-specific tastes, drugs that are relatively expensive in some locations can be instead relatively cheap for the in others without any changes in quality.More robustness to show that prices are driven less by cross-city variation and more by product quality, as per your assertion. Control for states as well as neighborhood characteristics. Or hold states and location constant by exploring price and quality within pharmacies within a spatial reference radius.Also, it is not clear with the design the characteristics of your silent shoppers: What are the exclusion criteria? Do you have their socio-economic attributes that can be used as a vector of covariates? For example, do they negotiate prices with pharmacies? Are they or a close family member on any medications similar to what they are shopping for? Do you randomly assign them to a location where they buy, or did you use the same location they are familiar with? More clarifications on these are needed.You can also control for store-level price indexes by obtaining the price of a commonly purchased commodity across Nigeria, e.g., sugar or salt, to determine whether prices are driven more by store characteristics than by drug quality.I'd also love to know how seasonal fluctuations affect demand, which can impact price. Was the survey conducted in the same month across the selected states? What are the indications of temperature and humidity that can affect disease outbreaks and subsequently drug demand and price across time and states?

We look forward to receiving your revised manuscript.

Kind regards,

Jubril Animashaun

Academic Editor

PLOS ONE

Journal Requirements:

“The project was funded by USAID-DIV (7200AA21FA00006) (EM) and internal funding from the University of Michigan (EM). The funders did not play any role in the study design, data collection and analysis, decision to publish, or preparation of the manuscript.”

“The project was funded by USAID-DIV (7200AA21FA00006) (EM) and internal funding from the University of Michigan (EM). The funders did not play any role in the study design, data collection and analysis, decision to publish, or preparation of the manuscript.”

4. We note that Figure S1 Fig 1 in your submission contain [map/satellite] images which may be copyrighted. All PLOS content is published under the Creative Commons Attribution License (CC BY 4.0), which means that the manuscript, images, and Supporting Information files will be freely available online, and any third party is permitted to access, download, copy, distribute, and use these materials in any way, even commercially, with proper attribution. For these reasons, we cannot publish previously copyrighted maps or satellite images created using proprietary data, such as Google software (Google Maps, Street View, and Earth). For more information, see our copyright guidelines: http://journals.plos.org/plosone/s/licenses-and-copyright.

a. You may seek permission from the original copyright holder of Figure S1 Fig 1 to publish the content specifically under the CC BY 4.0 license.  

Reviewers' comments:

Reviewer's Responses to Questions

**Comments to the Author**

1. Is the manuscript technically sound, and do the data support the conclusions?

Reviewer #1: Yes

Reviewer #2: Yes

2. Has the statistical analysis been performed appropriately and rigorously? 

Reviewer #1: Yes

Reviewer #2: Yes

3. Have the authors made all data underlying the findings in their manuscript fully available?

Reviewer #1: Yes

Reviewer #2: Yes

4. Is the manuscript presented in an intelligible fashion and written in standard English?

Reviewer #1: Yes

Reviewer #2: Yes

5. Review Comments to the Author

Reviewer #1: Dear authors,

Thank you for addressing most of the original comments. Please read through and check minor typos and grammar in few instances, e.g.

Line 98-100: Write in past tense

Line 180-182: This should be written in past tense.

Reviewer #2: Review for PLOS One:

Title: Is Price Associated with the Quality of Medicines? Quantitative Evidence from Nigeria

General: The study attempts to correlate price with medicine quality which is often a complicated relationship exists in reality. It was written well with good logical flow.

Major comments;

1. Quality is assessed in terms of HPLC assay results however medicine quality is assessed in terms of many parameters based on the dosage forms. It includes but not limited to Assay, dissolution, disintegrations, tablet hardness, friability, Limit tests, endotoxin tests etc… . Thus better to take this fact in to consideration in the title too…..Hence rather than using the broader term medicine quality in the title and other places….Please use “API content” or assay value….

2. What is the accreditation status of the testing lab (Hydrochrom Analytical Services Limited, a private laboratory located in Gowon Estate, Lagos). Better to include a statement about the accreditation status and quality assurance mechanisms’ deployed in the quality testing system. Hence the laboratory test results generated are reliable.

3. Price is presented in Nigerian Currency, Naira, only in table 1 and other figures. It is better if it is converted to more international currency like the US dollar for easy grasp by international readers.

4. Better to employ appropriate statistical tests for the statement of significance about the issue mentioned in lines 243-244 “Overall, the average price of drug samples that did not fail the laboratory test (NGN 1620) slightly higher than that of those samples that failed (NGN 1530)”.

5. The detailed analyses still did not demonstrate price and quality direct relationships ( increase in price ..good quality medicines) as mentioned in different places throughout the text… as seen in lines 275-281. Hence better to include such observations in the abstract and the conclusions thus readers can grasp the relations between prices and quality are not direct…….but also variable among therapeutic categories….

“The bimodal shape of the distribution is more pronounced for antihypertensives and antimalarials. In contrast, the distributions for analgesics and antibiotics are more spread out, suggesting that price may be more strongly associated with the quality of these two categories of medicines than the quality of antihypertensives and antimalarials. We also observe a similar pattern when plotting the percentage of samples by price bins, for each category of medicines (S1 Fig 3). Among analgesics, the failure rate is concentrated in the lowest price bin (“<ngn 1200=" ">Bins”.

6. Nigerian NAFDAC is at Maturity Level III (ML3) by WHO…… it implies the regulatory authority is mature enough to protect public health in Nigerian market. But ML3 is not achieved in many sub-Saharan African countries. Hence some of the descriptions that are mentioned in the text like “regulatory oversight is limited, infrastructure for quality testing is often inadequate”. These statements might apply in general context of Africa but might not be in Nigeria based on regulatory Maturity level….. Thus it either better to mention the NAFDAC’s Maturity level or contextualize the bold statements mentioned throughout the manuscript based on WHO evaluation bench marking mechanism of the Nigerian NAFDAC.

7. Why medicine organization by brand is emphasized in this study? It is known that medicines in pharmacy can also be organized by therapeutic class or alphabetical etc?

8. Although direct sun light, cold chain and other specialized storage conditions are mentioned the chemistry and storage requirements of the selected 20 medicines is not described in the manuscript in order to grasp the relationships….

9. The medicines used for laboratory data as well as in the price analyses in this manuscript are 20 brand products which are produced by specific manufacturers…. Hence as such, it will limit the purchase of different products available in each pharmacy. Hence I assume products other than these 20 brands will not be purchased. Based to include such limitations in the discussion too….

Minor:

-Why one product only is purchased from a pharmacy</ngn>

6. PLOS authors have the option to publish the peer review history of their article (what does this mean? ). If published, this will include your full peer review and any attached files.

**Do you want your identity to be public for this peer review?** For information about this choice, including consent withdrawal, please see our Privacy Policy .

Reviewer #1: **Yes: ** Dr Felix Khuluza, Pharmacy Department-Kamuzu University of Health Sciences, Malawi

Reviewer #2: No

---

## [Author Response · Author response to Decision Letter 1]

11 Nov 2025

Dear Prof. Animashaun,

We would like to thank you and the reviewers for the time and effort dedicated to evaluating our manuscript. We deeply appreciate the constructive feedback and insightful comments, which have helped us enhance the clarity, rigor, and overall presentation of our work.

In response to the revise and resubmit request, we provide a point-by-point response to each comment raised by the editor and reviewers in the response to reviewers letter. For each point, we indicate how we addressed it in the revised manuscript. All changes made in response to these comments are highlighted in the version with tracked changes. We have also uploaded an unmarked version of our revised paper without tracked changes.

We are grateful for the opportunity to revise and resubmit our manuscript, and we thank you for your thoughtful guidance throughout this process.

---

## [Editor Report · Decision Letter 1]

26 Nov 2025

Is Price Associated with the Quality of Medicines? Evidence from Active Pharmaceutical Ingredient Testing in Nigeria

PONE-D-25-41093R1

Dear Dr. Marie Chantel Montás,

We’re pleased to inform you that your manuscript has been judged scientifically suitable for publication and will be formally accepted for publication once it meets all outstanding technical requirements.

Kind regards,

Jubril Animashaun

Academic Editor

PLOS ONE

---

## [Editor Report · Acceptance letter]

PONE-D-25-41093R1

PLOS One

Dear Dr. Montás,

I'm pleased to inform you that your manuscript has been deemed suitable for publication in PLOS One. Congratulations! Your manuscript is now being handed over to our production team.

Kind regards,

on behalf of

Dr. Jubril Animashaun

Academic Editor

PLOS One